# Human Cutaneous Leishmaniasis in North Africa and Its Threats to Public Health: A Statistical Study Focused on Djelfa (Algeria)

**DOI:** 10.3390/microorganisms11102608

**Published:** 2023-10-22

**Authors:** Fatma Messaoudene, Slimane Boukraa, Said Chaouki Boubidi, Ahlem Guerzou, Abdeldjalil Ouahabi

**Affiliations:** 1Exploration and Valorization of Steppe Ecosystems Laboratory, Faculty of Nature and Life Science, Ziane Achour University of Djelfa, Djelfa 17000, Algeria; 2Department of Agricultural and Forestry Zoology, Ecole Nationale Supérieure Agronomique, El-Harrach 16004, Algeria; 3Laboratoire d’Eco-Epidémiologie Parasitaire et Génétique des Populations, Institut Pasteur d’Algérie, Dely-Brahim 16047, Algeria; 4UMR 1253, iBrain, Inserm, Université de Tours, 37020 Tours, France

**Keywords:** cutaneous leishmaniasis, epidemiology, risk factors, Djelfa, Algeria

## Abstract

Cutaneous leishmaniasis, the most common form of leishmaniasis, causes long-term skin lesions on exposed areas of the skin. It is caused by a protozoan parasite belonging to the genus *Leishmania* and is transmitted via infected phlebotomine sand flies. In North Africa, particularly Algeria, the disease represents a major public health problem. This retrospective study, which focuses on the agropastoral region of Djelfa (central Algeria) during a period of 16 years, from 2006 to 2021, is part of the surveillance of cutaneous leishmaniasis to identify the key factors favouring its probable spread. The analyzed data reveal that this disease is more prevalent in male patients (53.60%) and is highly widespread in this vast area of 66,415 km^2^ with a total of 3864 CL cases, reaching a peak of 1407 cases in 2006. Statistically, the Pearson correlation validated by the *p*-value shows, in an original and sometimes unexpected way, that certain factors, such as temperature linked to climate change, are playing a significant role in the probable spread of the disease in Djelfa and its surrounding regions. The concentration of the population in some specific rural areas with limited or nonexistent access to public health services is another potential factor in disease transmission. The results were highlighted by a significant correlation coefficient (r=0.66) with a *p*-value less than 0.01. While there is currently no vaccine or prophylactic drug available, our research represents a preliminary approach that addresses various epidemiological aspects of the disease. This paves the way for a proactive preventive strategy involving the control of vector-borne diseases.

## 1. Introduction

Leishmaniasis is a vector-borne disease caused by flagellated protozoan parasites belonging to the genus *Leishmania* Ross, 1903 (Kinetoplastida: Trypanosomatidae). Many mammalian species, including humans, can be affected by this parasite. The disease is endemic in several tropical and subtropical countries and regions on many continents [1,2]. It manifests in several forms in humans including mucocutaneous, visceral, post-kala-azar dermal and diffuse cutaneous and cutaneous leishmaniasis (CL) [3,4]. In fact, this group of diseases comprises more than 20 *Leishmania* spp. pathogenic to humans, which can be transmitted by approximately one hundred known vectors or suspected sand fly species [5,6]. The disease form (CL) investigated in this paper is characterized by cutaneous lesions, mainly open sores on exposed skin, causing long-term skin damages, with 1.5 million new cases per year [7,8].

In North Africa, CL is common in several countries, including Tunisia, Morocco, Libya and Algeria. In the latter country, Biskra is well known as an important focus of this disease, in which the first case was recorded back in 1860 [9,10]. Subsequently, it expanded to create large outbreaks of infection, creating a serious public health problem [11,12,13].

The proven reservoir hosts for *Leishmania* spp. are rodents (*Psammomys obesus*, *Meriones shawii* and *Meriones libycus*), and the suspected ones are canines (*Canis* spp.) [10,14,15,16,17,18]. Four endemic *Leishmania* species are recorded, namely *L. major*, *L*. *tropica, L. killicki* and *L*. *infantum,* which are distributed in a variety of bioclimatic stages along with a diversity of vectors and reservoir hosts, given that each of these species has different epidemiological and clinical characteristics [19,20,21]. Whereas *L. major* is widespread in arid and Saharan zones; *L. tropica*, as well as *L. killicki*, is known to occur in almost all type of bioclimates, and *L. infantum* is found in humid, sub-humid and semi-arid zones [10]. These pathogens, which are in their amastigote form, are ingested via the bite of sand flies belonging to the *Phlebotomus* genus (Diptera: Psychodidae), including *P*. *papatasi*, *P*. *sergenti*, *P*. *perfiliewi*, *P*. *perniciosus, P. ariasi* and *P*. *longicuspis* [21,22,23,24,25,26]. In their gut, amastigotes transform into a flagellate promastigote in 24–48 h and are divided via binary division before migrating to the pharynx. The infection happens when a phlebotomine sand fly bites and injects promastigotes directly in host humans. The latter transform back into amastigotes and multiply within the mononuclear phagocytic cells. After the rupture of the host cell, multiple amastigotes release into the blood and remain locally concentrated in the skin tissue, leading to an ulceration on the skin [27,28,29].

In our study area, Djelfa, CL surveys are fragmentary and poor with only few reports from health services despite the fact that the region has many factors favouring the probable development of this disease. Indeed, Djelfa borders eight provinces, including Biskra, a former focus of leishmaniasis that is still active (24,232 cases over a ten-year period); M’sila, which is one of the major CL-infected regions in recent years, with 3000 cases per year; and Ghardaia, a known focus of various zoonoses in Algeria, particularly leishmaniasis, with 9328 cases over the last twenty years [11,30,31].

The agropastoral vocation of the region, with its intense sheep-rearing activity, contributes greatly to the nourishment of leishmaniasis vectors [32,33] and favours the presence of rodents, in particular *Meriones shawii*, the main reservoir of *leishmania*. In addition to all these factors, the climate of Djelfa is characterized by a semi-arid climate with rainy, cold winters and hot, dry summers, which favours the development of sand flies, the vectors of *leishmania*.

To this end, our study aims to complete the following:Survey the situation of CL in the Djelfa province over a period of 16 years and identify potential risk factors, such as climatic parameters that facilitate the spread of the disease;Identify specific strains of *Leishmania* since different regions may harbour different *Leishmania* species. This study will help to identify the specific strains present in Djelfa and to understand the genetic diversity of the parasite, which is crucial for the development of control strategies;Determine the potential influence of local factors, such as geography, climate, ecology and human behaviour, on the spread of the disease by focusing on Djelfa. In order to explore how these specific local factors may constitute risks for the transmission of cutaneous leishmaniasis, this study takes into account that these factors may differ from those in other regions of Algeria and even other North African countries;Clarify the impact of cutaneous leishmaniasis on the local population by understanding its extent in the region so that public health authorities will be able to allocate resources more appropriately for prevention, screening and treatment;Address the local concerns of Djelfa residents who may have specific concerns about the incidence of cutaneous leishmaniasis in their region in order to respond directly to these concerns, leading to a greater social acceptance of prevention and control measures.

## 2. Materials and Methods

### 2.1. The Study Area

Djelfa province (2° to 5° E and 33° to 35° N) represents a pastoral area, where livestock rearing is the main activity [34]. The province is situated amidst the steppe in the central part of Algeria, covering an area of 66,415 km^2^ with a population of 1,538,476 (according to the latest 2018 census) distributed over 36 communes. It is delimited by eight provinces, of which Biskra and M’sila in the east and Ghardaïa in the south, respectively, represent the regions most severely affected by CL in Algeria [30,31,35] (Figure 1).

### 2.2. Data Collection

Five local public health institutes—Ain Oussera and Hassi Bahbah in the north, Djelfa in the center and Messaad and Guettara in the south (Figure 1)—are linked to the Department of Health and Population, where monthly reports are circulated throughout the year. Epidemiological data collected for this study are directly provided by the Department of Health and Population of Djelfa. Declared patients and population density in 36 communes were provided between 2006 and 2021, although the information according to gender and age groups in each commune is only available from 2009 to 2021. The climate data used in this study (temperature and precipitation) were obtained from the National Meteorological Office.

### 2.3. Diagnostic Investigations

In the study region, patients were diagnosed in hospitals and dermatological clinics. Investigative tests are performed in two different ways: firstly, via microscopic examination with Giemsa-stained swabs taken from biopsy specimens, infiltrated lesions and ulcer margins, and secondly, they were taken via molecular diagnosis in complex cases, due to the efficiency and sensitivity of polymerase chain reactions [36,37].

### 2.4. Data Mapping

ArcGIS 10.7 software was applied to map the geographical spread of the CL disease in the study region. Shapefiles of the boundaries were uploaded from DIVA-GIS (https://www.diva-gis.org/gdata) (accessed on 22 January 2022).

### 2.5. Statistical Analysis Tools

Typically, the fundamental principles of advanced statistics are applied to biological processes. Based on this, the Pearson correlation between CL cases and potential factors such as precipitation, temperature variables (minimum, maximum and mean) and population density was analyzed and interpreted using the IBM SPSS Statistics 20.0 software for further processing [38,39].

## 3. Results

### 3.1. Annual Distribution of Cutaneous Leishmaniasis Cases in Djelfa

A total of 3864 CL cases were declared between 2006 and 2021, with different annual variations. The peak reached 1407 cases in 2006. Thereafter, the incidence declined sharply, reaching 33 cases in 2009 and then rising gradually to 255 cases in 2011, followed by a drop to 34 cases in 2014 and a rise to 445 cases in 2018, continuing to fluctuate between increases and decreases over the last three years (2019, 2020 and 2021) of the study (Table 1).

### 3.2. Monthly Variations of Cutaneous Leishmaniasis Cases in Djelfa Province (2009–2021)

Regarding seasonal variations in all periods of the study, the percentage of cases of CL was very high during the winter period, with 23% in January and 16% in February, then it decreased to a minimum during the summer period, from 1% in July to a gradual rise over the months until it reached 12% of cases in November and 20% in December (Figure 2).

### 3.3. Distribution of Cutaneous Leishmaniasis Cases According to Age Groups and Gender (2009–2021)

The proportion of cutaneous leishmaniasis cases based on gender shows a male predominance with 1213 cases (53.60%). In comparison, females came second with 1050 cases, representing 46.40% of all patients. According to patient age, the highest incidence of CL is found among the 20-44 age group with 34.87%, followed by the 10–19 age group with 19.22%. The incidence rate among patients aged less than one year was weak at 2.34% (Table 2).

### 3.4. Distribution of Cutaneous Leishmaniasis Cases by Commune in Djelfa Province (2006–2021)

A breakdown of cutaneous leishmaniasis cases per commune in Djelfa province reveals that the highest recorded case rate is in the Messaad commune (807 cases; 20.89%), followed by Ain Oussera (797 cases; 20.63%), Hassi Bahbah (451 cases; 11.67%) and Djelfa (275 cases; 7.12%). The lowest CL incidence was declared in the communes of Boughezoul, Idrissia and Douis (one case; 0.03%). A map illustrating the distribution of CL in the various communes of Djelfa is shown on the next page (Figure 3).

### 3.5. Analysis and Interpretation of the Findings

The dependent variable (Y) is the 12-month average of CL incidence between 2009 and 2021. The independent variables (X) are the population density data for 35 communes and the 12-month average of all relative temperatures (minimum, maximum and mean) and precipitation from 2009 to 2021.

Climatic data and population density were introduced into the IBM SPSS Statistics 20 software for further analysis. Pearson’s correlation between CL cases and potential factors revealed the following:-With the populated density, there is a very strong presumption against the null hypothesis $\left(p<0.01\right)\;.$ The two variables are strongly positively related (r=0.686). Among the 36 communes surveyed, Djelfa (533.5 people/km^2^), Ain Oussera (125.1 people/km^2^), Hassi Bahbah (111.7 people/km^2^) and Messaad (693.4 people/km^2^) are the most populous in the Djelfa province (Figure 4);-With the temperature minimum, there is a very strong presumption against the null hypothesis $\left(p<0.01\right).\;$The r is so close to −1. The two variables are strongly negatively related (r=−0.876). Tmin was ranging between 1 °C in January and 20.7 °C in July;-With the temperature maximum and mean, there is a strong presumption against the null hypothesis (p<0.05). The two variables are moderately negatively related, and the r of the two relatives is, respectively, (r=−0.594 and r=−0.590). The temperature trends showed that the Tmax varied from 9.7 in January to 35 in July, while the monthly mean temperature ranged from 5.3 °C in January to 35.8 °C;-With the precipitation, there is a weak presumption against the null hypothesis (p>0.05). The two variables are weakly positively related (r=0.331). The range of precipitation was 32.7 mm in September, and the lowest value was 7.7 mm in July (Figure 5).

## 4. Discussion

The surveillance of cutaneous leishmaniasis in the agropastoral region of Djelfa in central Algeria over a 16-year period (2006-2021) revealed a total of 3864 cases.

Given Djelfa’s vocation for extensive sheep farming [40] and the presence of rodents, notably *Meriones shawii* that acts as a reservoir host for cutaneous leishmaniasis [16,41], our results underline the region’s inherent susceptibility to cutaneous leishmaniasis.

Our research highlighted that, during the study period, the scale of infection reached its peak in 2006, when an impressive number of 1407 cases of CL were reported. Prior to this date, it is likely that cases of infection were reaching unprecedented levels. However, since the public authorities began recording cases in 2006, there has been a decline, likely attributable to a greater awareness of the seriousness of the situation and the implementation of preventive measures. 

This leads to the consideration of the hypothesis that urban expansion into previously uninhabited areas, such as the oueds (in North Africa, an “oued” is a term used to describe a dry riverbed or seasonal river. Oueds are typically river valleys or channels that only contain water during the rainy season or after heavy rainfall but are dry for most of the year) around the city that year, led to the population coming into direct contact with a new territory that was already inhabited by sand flies and rodents. This, in turn, prompted the government to implement a sand fly control program, ultimately resulting in a decrease in case rates [42]. This was the situation in our research, where CL cases gradually declined to 33 cases in 2009, then fluctuated with increases and decreases, justifiable to some extent by the excessive use of insecticides in the health sector as well as on farms, which contributed to the development of insecticide resistance in the disease vectors [43]. Additionally, inadequate sanitation and irregular household garbage disposal, combined with the construction of rural houses from mud with wall cracks, facilitate the breeding of phlebotomine sand flies [44,45]. The highest incidence of CL occurred in the 20-44 age group, with 34.87%, due to professional activities, in particular agricultural and livestock farming [46]. Indeed, gender-specific incidence rates of CL cases in Djelfa show a male predominance at 53.60%. This corresponds to the short survey carried out by Hamiroune et al, where men were strongly affected, with 57.83%, compared to women across 21 communes in Djelfa [47]. 

Epidemiological analyses carried out on a monthly basis showed that in winter, the highest number of CL cases was reported in January (23%), February (16%), November (12%) and December (20%), then decreased to a minimum during summer (1% in July). These findings are in line with those reported in Sudan (East Africa), where the greatest incidence was noted during the winter period, with approximately 450 cases, followed by a progressive decrease to an estimation of 90 cases in the summer [48]. This can be explained by the fact that the incubation time of *Leishmania* spp. in the human host is, on average, 1 to 6 months, and a diagnosis is usually completed during autumn and winter, given that contamination mainly occurs in the late summer, due to the activity of sand flies—the main vectors of *Leishmania* spp. during the summer—since temperature is proven to significantly affect the development time and metabolic activity of these vectors [49,50].

The distribution of CL cases in the study area has revealed that the highest incidences are reported in the municipalities with well-equipped local public healthcare facilities, such as Messaad in the south and Djelfa, Hassi Bahbah and Ain Oussera in the north. In fact, these municipalities also account for CL-infected patients from the neighbouring municipalities that lack healthcare facilities.

The risk of emergence or re-emergence of vector-borne Zoonoses may be influenced by ecological changes such as extreme weather conditions (temperature and precipitation), which will impact the spread of both vectorial and infectious diseases [51]. Several surveys revealed associations between CL and climate factors in various regions, leading to the conclusion that all these climatic parameters can influence vectors and vector-borne diseases, potentially favouring the spread of this epidemiology [50,52,53,54]. It is important to highlight the fact that temperature significantly influences the density and dynamics of the vectors and reservoirs of this disease. This is consistent with findings reported in Iran and Colombia, where temperature seasonality correlated negatively with CL [52,53]. Nevertheless, in central Tunisia, no significant correlation was observed between temperature and CL cases [54], while in the province of Djelfa, a non-linear correlation with a weak positive relationship was noted, which can be explained by the fact that heavy rainfall likely destroys phlebotomine sand fly breeding sites [50]. It should be noted that the province of Djelfa is well known for its semi-arid climate, characterized by very long dry periods, followed by heavy rainfalls. Furthermore, a strong, positive correlation has been shown between CL cases and population density. The same trend is valid for other tropical diseases [55]. Fast population growth and unplanned urbanization are major risk factors, which might be linked to population migration to urban areas, moving from villages to poorer parts of cities [56]. Importantly, the disease is prevalent in rural areas opening up to urban areas [57]. Sub-standard housing and primitive sanitary conditions in these low-income societies are responsible for this situation [58]. In addition, the uncontrolled urbanization of areas without respect for health and safety standards is affecting economic and social development and increasing poverty [59]. Consequently, the risk of malnutrition is ongoing, and immune systems are compromised, leading to greater susceptibility to a CL infection [60]. Delay in treatment due to high costs of drugs leads to disease exacerbation, coupled with an increase in disease incidence [61].

In North African countries, cutaneous leishmaniasis (CL) represents one of the most frequent parasitic diseases, threatening public health [62,63,64,65,66]. Since the 1980s, CL incidence has spread in North African countries, mainly in Algeria, where cases range from 10,000 to 40,000 per year [1,10], making this country among the most affected by CL. This vector-borne disease is reported in several regions from the north to south and east to west of Algeria [12,20,37,47,67,68,69]. 

In Tunisia, the annual number of CL cases was 669.7/100,000 inhabitants for six more years, with an estimated 15,897 cases reported in urban and rural areas in 2004 [70]. Due to their work in fields and pastures and their close proximity to livestock, women have been the most affected by this disease [71,72]. It is important to note that in Tunisia, women have the opportunity to pursue any occupation thanks to one of the most progressive legislations for women in North Africa.

The same pattern of incidence of CL is reported in Morocco, where women and children were highly affected by the disease, with the average number of cases over ten years reaching 139/100,000 in both rural and urban areas [73,74], reflecting urbanization, which positively impacted the number of leishmaniasis cases, whilst poverty showed no effect on the incidence [75]. 

These findings are in line with surveys conducted in Libya, where housewives and children were most affected by CL in rural areas due to agricultural activities, which increased the risk of infection to 3000 cases in 2006 [76,77,78]. 

Conversely, men in rural areas were highly affected due to their work in agricultural fields in the Djelfa study region [79]. This leads to the hypothesis that agricultural activity (in particular sheep breeding) exposes people to the phlebotomine sand flies responsible for the transmission of *Leishmania* spp. Moreover, this social activity may justify the high rate of infection among men in some areas and in women in others. Furthermore, the predominance of males is linked to the high association between the immune–endocrine interaction and CL incidence, which plays a critical role in the progression of disease [80]. 

According to the World Health Organization, CL is highly endemic in North Africa, where the countries with the highest incidence—affecting all patients of different genders, ages and socio-economic backgrounds—are, respectively, Algeria (7051 cases; 47.32%), followed by Morocco (3141 cases; 21.08%), Tunisia (2732 cases; 18.33%), Libya (1095 cases; 7.35%) and Egypt (883 cases; 5.93%) [81]. However, despite these significant figures, the World Health Organization (WHO) points out that for many years, the impact of leishmaniasis on public health worldwide has been greatly underestimated [2]. This spread of the disease is likely due to climatic conditions, human activity and the movement of populations in endemic areas, which favour the reproductive activity of phlebotomine sand fly vectors, so specific control measures are needed to reduce the populations of phlebotomine sand flies and rodents by improving general health and hygiene conditions, which in turn could reduce the incidence of this disease [82].

In order to automatically monitor the propagation of cutaneous leishmaniasis in the agropastoral region of Djelfa and beyond, and to allow an accurate detection and identification of the disease from images, the implementation of tools from artificial intelligence is necessary [83].

Indeed, since we have a large database, it is possible to use the concept of Deep Learning, based on an architecture of the convolutional neural network types usually used in biometrics and computer vision [84,85,86,87,88,89]. In the case where the images are noisy, it is possible to perform a denoising using a simple but quasi-optimal approach based on wavelets [90,91,92] or a denoising based on the concept of compressed sensing, which has recently shown its good performance [93]. Such tools will allow us to objectively evaluate the performance of our future approach (work in progress) using metrics such as accuracy, precision and F1 score [94,95].

## 5. Conclusions

The evolution of cutaneous leishmaniasis in the Djelfa province confirms that the disease is endemic and represents a major public health issue in all the communes prospected. From 2006 to 2021, a significant annual incidence rate was recorded, with often alarming figures, peaking in 2006. The incidence was reported for all genders, with a clear predominance of men aged between 20 and 44, due to their intense activity in agropastoral activities. The Pearson correlation test showed that CL incidence is negatively correlated to relative temperatures, weakly positively correlated to rainfall and strongly positively correlated to population density. The agropastoral vocation of Djelfa offers a favourable environment for the spread of phlebotomine vectors and potential reservoir rodents. Preventive measures for controlling the spread of the disease involve the control of adult vectors, the destruction of phlebotomine sand fly breeding sites and the elimination of chenopods, under which rodents feed and build their burrows. On a global scale, research on effective vaccines and drugs, as well as diagnostic tools, must be promoted. These measures will be followed by tasks of detection and identification of disease, immediately using images taken with a simple smartphone, which will deliver an instant diagnosis based on artificial intelligence and assess the confidence (probability of error, accuracy and precision) given to our diagnosis.

## Figures and Tables

**Figure 1 microorganisms-11-02608-f001:**
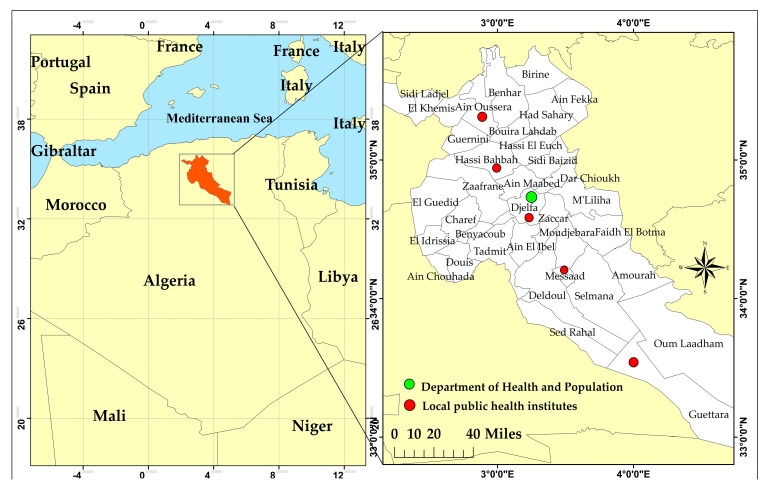
Geographical location of Djelfa province showing its communes and public health institutions.

**Figure 2 microorganisms-11-02608-f002:**
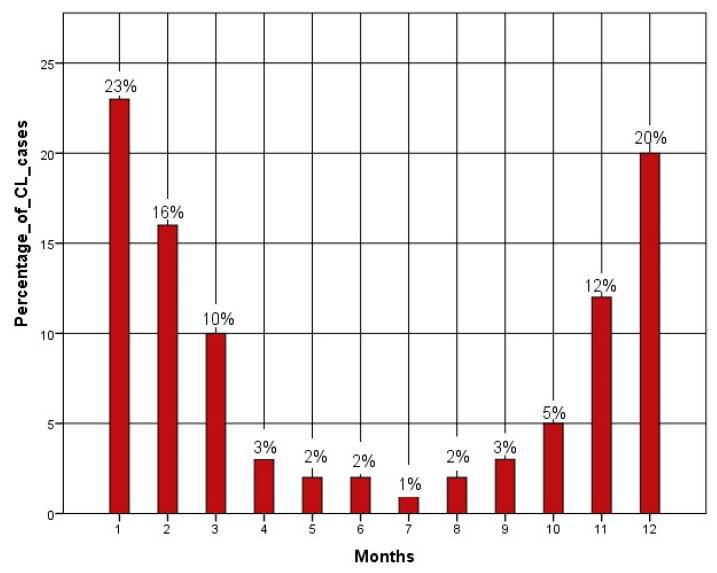
Monthly variation of CL cases in Djelfa province (2009–2021).

**Figure 3 microorganisms-11-02608-f003:**
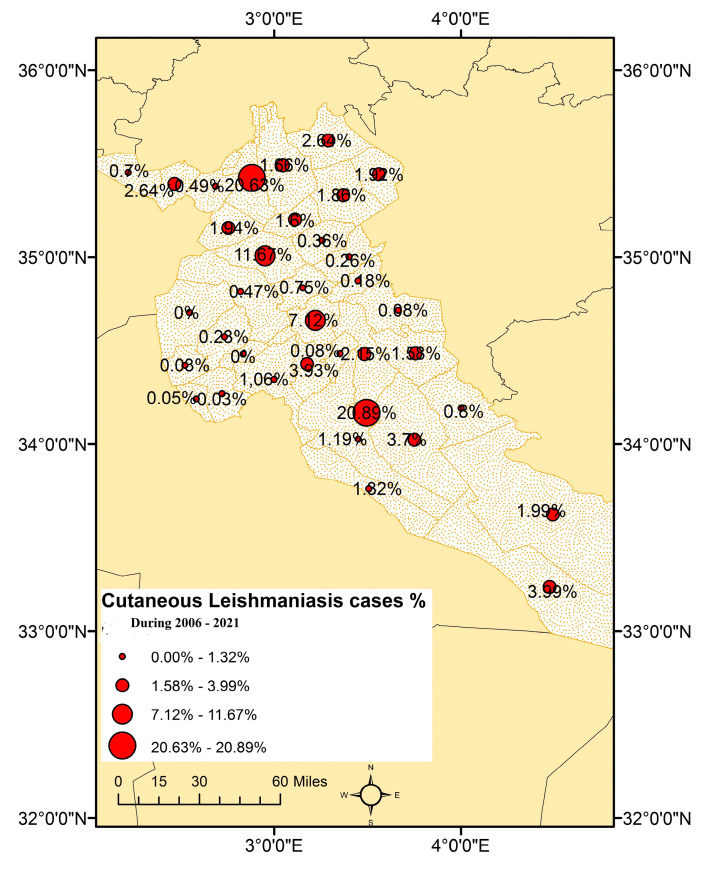
Spatial distribution of CL cases by commune in Djelfa province. Communes with weak or strong incidence are represented by circles, and CL cases are indicated by percentages (%).

**Figure 4 microorganisms-11-02608-f004:**
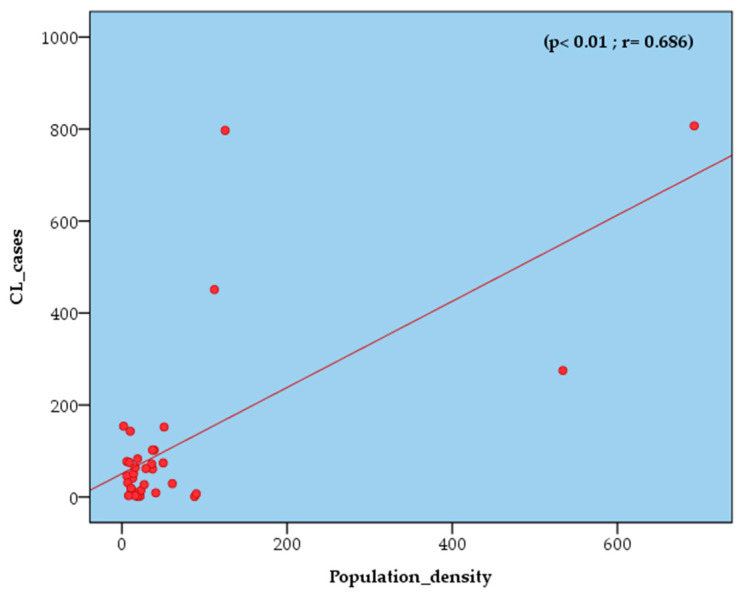
Pearson correlation: between annual CL incidences from 2006 to 2021 vs. population density.

**Figure 5 microorganisms-11-02608-f005:**
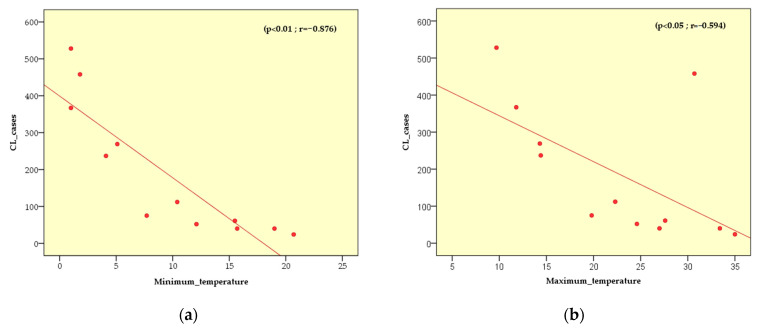
Pearson correlation: between CL incidences from 2009 to 2021 vs. climate factors—(**a**) minimum temperature; (**b**) maximum temperature; (**c**) mean temperature and (**d**) precipitation. The correlation is significant at the 0.01 level (p<0.01) and at the 0.05 level (p<0.05).

**Table 1 microorganisms-11-02608-t001:** Evolution of the annual cases of CL in Djelfa province (2006–2021) (data-source: Department of Health and Population).

Evolution of the Annual Cases of CL in Djelfa Province (2006–2021)																	
Years	2006	2007	2008	2009	2010	2011	2012	2013	2014	2015	2016	2017	2018	2019	2020	2021	Total
CL cases	1407	154	40	33	49	255	210	44	34	45	50	222	445	251	354	271	3864

**Table 2 microorganisms-11-02608-t002:** Distribution of CL cases based on age groups and gender in Djelfa (2009–2021).

Age Groups	Gender	Number of Cases	Percentage %	Total
00–01 year	M	31	1.37%	2.34%
	F	22	0.97%	
02–04 years	M	125	5.52%	9.59%
	F	92	4.07%	
05–09 years	M	153	6.76%	12.15%
	F	122	5.39%	
10–19 years	M	227	10.03%	19.22%
	F	208	9.19%	
20–44 years	M	430	19.00%	34.87%
	F	359	15.86%	
45–64 years	M	175	7.73%	16.39%
	F	196	8.66%	
Over 65 years	M	72	3.18%	5.44%
	F	51	2.25%	
Total	M	1213	53.60%	100 %
	F	1050	46.40%	

M: Male; F: Female.

## Data Availability

Not applicable.

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
