# Peer review of "Human Cutaneous Leishmaniasis in North Africa and Its Threats to Public Health: A Statistical Study Focused on Djelfa (Algeria)"

_microorganisms, 2023, doi:10.3390/microorganisms11102608_

Round 1

Reviewer 1 Report (Previous Reviewer 2)

The authors have addressed all my comment and accepted almost all of my suggestions. From my side, I do not think that further revisions are required. The authors have addressed all my comment and accepted almost all of my suggestions. From my side, I do not think that further revisions are required.

Author Response

Thank you for your thorough review of our paper and for accepting it.

Reviewer 2 Report (New Reviewer)

Dear authors,

after reading and partly correcting your manuscript "Human Cutaneous Leishmaniasis in North Africa and its Threats to Public Health: A Statistical Study Focused on Djelfa (Algeria)" I have to state that my suggestion is to reject your manuscript. Here are only some of my concerns (the rest of them are written in corrected version of the manuscript):

1) There are many wrong citations in introduction- please correct them (also I didn't check the whole paper's references because you missed to many of them up to the end of the Introduction section). Please be so kind and before submitting in other journal check your references and correct them as the wrong citations are very misleading for the reviewers/readers.

2) You should ask for help of English lector or native speaker to correct your language.

Best regards

You should ask for help of English lector or native speaker to correct your language.

Author Response

Thank you for your thorough review of our paper. Please find below our responses given point-by-point.

Please also refer to the revised manuscript whose changes are highlighted in red.

Thank you for your valuable comments. We gratefully acknowledge your great contribution to maintaining the high  quality of the manuscript.

Reviewer 3 Report (New Reviewer)

Dear authors, thank you for the reliable work. The manuscript is very-well conducted and wrote. Below, we make few comments about some points that could be clarifier to the readers.

·         The Introduction section have good ideas and well-made revisions, but the construction is quite confused, with ideas dispersed and redundant information. Please, re-wrote the introduction.

o   For example: The first paragraph is like a mini-introduction of the all section. You could insert each detailed information together these initial phrases: when talk about sandfly transmission, cite the different species, removing from the next paragraph, thus avoid repetition.

·         You could include in the justification (last paragraph of the Introduction) what the importance of these types of studies not only for Dejfa, but also for all scientists from the field and other persons (as to correct governmental interventions or the better understanding dynamics of transmission and impacts of climate changes).

·         The data was obtained directly from “national data” or from institutes directly? Please, provide more information about data collection, type of study and others.

·         Why so many cases in 2006? Was also the first year of diagnostic surveillance of CL? You need to discuss this in Discussion section.

·         I really disagree of this information: “Since 2006, the number of affected subjects has decreased considerably to reach 33 cases in 2009 and gradually increase again to reach 354 cases in 2020 (Table 1).” – We see four years of decreased cases numbers. And a normalized and increased number of cases since 2017 until now. Please, rewrote the description of these data.

·         The Discussion section need many improvements:

o   Many papers indicate that higher incidence of leishmaniasis in males could be related to testosterone presence and not only related to the behaviour and labour activity;

o   Please, make more concise text. There excess of comparison that does not lead to improved comprehension of data, but few discussions of your findings per se;

o   As indicated above, the higher number of cases in the year of 2006 need be clarified.

Minor comments:

Line 36: Please, use phlebotomine (sandfly is colloquially);

Line 44: The use of italic in leishmania is unnecessary (authors could change to leishmania or Leishmania (preferred));

Line 249: Leishmania spp. and not leishmania;

Suggestion: Maybe a revision by native anglophone or professional could improve the paper.

Author Response

Thank you for your thorough review of our paper. Please find below our responses given point-by-point.

Please also refer to the revised manuscript whose changes are highlighted in red.

Thank you for your valuable comments. We gratefully acknowledge your great contribution to maintaining the high  quality of the manuscript.

Round 2

Reviewer 3 Report (New Reviewer)

Dear authors, thank you for the corrections in the manuscript. Including several updates in references and manuscript organization.

Please, before the submission of the final version: Once you choose write leishmania in italic, please, use the L letter in capital format, so using Leishmania and Leishmania spp. (in the paragraph from the lines 43-57).

Author Response

Thank you for your thorough review of our paper. Please find below our responses given point-by-point.

Please also refer to the revised manuscript whose changes are highlighted in red.

First of all, we thank the reviewer for these important remarks

This manuscript is a resubmission of an earlier submission. The following is a list of the peer review reports and author responses from that submission.

Round 1

Reviewer 1 Report

Dear Editor

Human Cutaneous Leishmaniasis in North Africa and its threats to Public Health: A Statistical Study Focused on Djelfa (Algeria) written by Messaoudene and colleagues

Although the subject of this article can be interesting but it is very poorly written suffering from serous lack of essential information listed in follow:

1.       The objective of this study is not clear. Is it a descriptive epidemiological study reporting the CL incidence in Djelfa or an analytical study to evaluate the probable correlation between incidence of CL and climate changes?

2.       There is no problematic in the introduction to express the actual status of leishmaniasis in studied area and to justify why this study was conducted.

3.       The text in particularly introduction is very long with several unrelated general text without any correlation with subject of this study.

4.       The hypothesis stated in materials and methods does not explain the authors really would evaluate the correlation of which elements?

5.       The incidence of CL in an endemic area is not only affected by ecological or weather events (temperature and precipitation), but also by geographical positions and barriers, humidity, altitude, vegetation, reservoir fauna, etc. Why the authors only considered the temperature in their statistical analysis.

I proposed some remarks in the introduction and abstract sections (pdf attached) but due to the lack of some essential information mentioned above, it is difficult to read the following text without understanding the goal and essential points of article.

Reviewer 2 Report

English still needs to be reviewed, preferably by a native speaker of the language – this reviewer will not point out all the details requiring that approach

Line 14 – replace semicolon (;) with comma (,)

Line 15 – adapt to read as: It is caused by protozoa transmitted by infected female phlebotomine sand flies.

Line 17 – (northern Algeria)

Line 18 – the period of 2006 to 2021

Line 21 – define that p-value

Abstract – simplify the presentations of p-values

Keywords – display alphabetically

Line 33 – delete (L.)

Line 36 – sand fly

Line 41 – cutaneous leishmaniasis

Line 44 – Leishmania

Line 45 – not clear: Canis plural species

Line 46 – write out genus name (Leishmania) the first time each species is presented

Line 48 – insert comma after flagella,

Line 48 – sand flies [change accordingly throughout the manuscript] … to the genus Phlebotomus

Line 53 – replace reticuloendothelial with mononuclear phagocytic

Line 60 – CL

Table 2 – one decimal place would be enough for percentages

References – please standardize, e.g. titles should be presented in lowercase, as much as possible; Latin names should appear in italics (Genus species), etc.